# Management of a Patient with Tuberous Sclerosis with Urological Clinical Manifestations

**DOI:** 10.3390/medicina56080369

**Published:** 2020-07-23

**Authors:** Vlad Padureanu, Octavian Dragoescu, Victor Emanuel Stoenescu, Rodica Padureanu, Ionica Pirici, Radu Cristian Cimpeanu, Dop Dalia, Alexandru Radu Mihailovici, Paul Tomescu

**Affiliations:** 1Department of Internal Medicine, University of Medicine and Pharmacy Craiova, 200349 Craiova, Romania; vldpadureanu@yahoo.com; 2Department of Urology, University of Medicine and Pharmacy Craiova, 200349 Craiova, Romania; pdragoescu@yahoo.com (O.D.); victorstoenescu@yahoo.com (V.E.S.); paul.tomescu@yahoo.com (P.T.); 3Department of Biochemistry, University of Medicine and Pharmacy of Craiova, 200349 Craiova, Romania; 4Department of Anatomy, University of Medicine and Pharmacy of Craiova, 200349 Craiova, Romania; 5Student, University of Medicine and Pharmacy Craiova, 200349 Craiova, Romania; cimpeanu_r@yahoo.com; 6Department of Pediatrics, University of Medicine and Pharmacy of Craiova, 200349 Craiova, Romania; dalia_tastea@yahoo.com; 7Department of Cardiology, University of Medicine and Pharmacy Craiova, 200349 Craiova, Romania; drmihailovici@yahoo.com

**Keywords:** tuberous sclerosis, angiomyolipomatosis, uretero-hydronephrosis, angiofibromas

## Abstract

The tuberous sclerosis complex (TSC) is highly variable as far as its clinical presentation is concerned. For the implementation of appropriate medical surveillance and treatment, an accurate diagnosis is compulsory. TSC may affect the heart, skin, kidneys, central nervous system (epileptic seizures and nodular intracranial tumors—tubers), bones, eyes, lungs, blood vessels and the gastrointestinal tract. The aim of this paper is to report renal manifestations as first clinical signs suggestive of TSC diagnosis. A 20-year-old patient was initially investigated for hematuria, dysuria and colicky pain in the left lumbar region. The ultrasound examination of the kidney showed bilateral hyperechogenic kidney structures and pyelocalyceal dilatation, both suggestive of bilateral obstructive lithiasis, complicated by uretero-hydronephrosis. The computer tomography (CT) scan of the kidney showed irregular kidney margins layout, undifferentiated images between cortical and medullar structures, with non-homogenous round components, suggestive of kidney angiomyolipomas, bilateral renal cortical retention cysts, images of a calculous component in the right middle calyceal branches and a smaller one on the left side. The clinical manifestations and imaging findings (skull and abdominal and pelvis CT scans) sustained the diagnosis.

## 1. Introduction

The tuberous sclerosis complex (TSC) is an extremely variable disease that can affect any organs, from the brain, eyes, kidneys, skin, heart, and lungs, to occasionally the bones, because of growing benign tumors [1]. Until the 1980s, TSC was underdiagnosed when individuals with less severe manifestations of the disease began to be recognized. Nowadays, it has been established that the genetic component of this disease is linked to TSC1 and TSC2 genes’ alterations [2]. Pathogenic mutations in either the TSC2 gene at chromosome 16p13.3, or the TSC1 gene at chromosome 9q34, cause a multisystem disorder that greatly varies in extent and severity [3]. TSC has an autosomal dominant mode of inheritance, with almost complete penetrance, but variable expressivity [1]. Both genders and all ethnic groups could be affected by TSC [1]. More so, since 2010 the m-TOR inhibitor drugs are administrated as a medical treatment for the benign tumors, in combination with a possible surgical management [2].

## 2. Case Report

A 30-year-old Romanian patient, female, from urban background, with significant family history, came to the Emergency Department of our hospital for colicky pain in the left lumbar region, pollakiuria, hematuria, dysuria, nausea, unspecified vomiting, fever, chills and cough. All these symptoms had started 2 days prior to the presentation and had intensified in the last 3–4 h. The patient had a suggestive family history for neurological diseases (a grandfather with a kidney tumor, father with epilepsy, uncle (brother of father)—epilepsy and intellectual disability, aunt (sister of father)—intellectual disability, and a sister with epilepsy). At the admission the patient had a Glasgow Coma Scale score of 15, a respiratory rate of 16 rpm, 99% Oxygen Saturation, blood pressure of 135/95 mmHg, with a heart rate of 90 beats/minute, and body temperature of 37,7 °C.

The findings of the physical examination were: normal body build, pale skin and mucosae, facial angiofibromas on the nasal wings, cheek and chin (Figure 1), hypopigmented plaques on the lower limbs (Figure 2), confetti-like lesions on the legs (Figure 3), small shagreen spots on the posterior hemithorax (Figure 4), tachycardia, painful abdomen and highly sensitive left flank, left costovertebral angle tenderness, pollakiuria. Due to the presence of typical skin lesions TSC diagnosis was suspected.

Neurological examinations showed reduced osteotendinous reflexes (bicipital and patellar), without other pathologic findings. The ultrasound examination showed liver with nonhomogeneous echo pattern, steatosis, possible homogeneous hyperechoic hemangiomas. Kidneys appeared with irregular contours difficult to delimit from the surrounding tissues, very much modified structure with multiple, well delimited nodular images with mixed structure, consistent with ultrasound findings of angiomyolipomas and mild pyelocaliceal dilatation, both suggestive of bilateral obstructive lithiasis, complicated by uretero-hydronephrosis. The echocardiography examination showed aortic aneurysm and ECG showed sinus arrhythmia.

The chest X-ray results were normal, but the computer tomography (CT) scan of the abdomen and pelvis evidenced irregular kidney contour layout, undifferentiated images between cortical and medullar structures, with non-homogenous round components, suggestive of bilateral kidney angiomyolipomas, renal cortical retention cysts (with diameters between 3–16 mm), with dilatatory secretion in the right side and conserved secretion and excretion on the left side. Images of a lithiasic component were also found on the middle right caliceal branches (size of 7.1/6.5 mm) and one of the left side with a smaller dimension, 2.5/2.5 mm (Figure 5, Figure 6 and Figure 7). Similarly, the computer tomography scan of the abdomen and pelvis showed bilateral inflammatory infiltration in soft tissue surrounding the kidney and the entire surface of the ureters, as well as a small quantity of fluid in the right perirenal area—suggestive of pyelonephritis, periurethritis and angiomyolipomatosis. The computer tomography scan of the abdomen and pelvis revealed subcentimeter periaortic and intraortocaval lymph nodes. Moreover, the CT changes were represented by the evidence of the osteolytic lesions in the thoraco-lumbar and pelvic bones. The CT scan of the skull revealed the following changes: a subependymal giant cell astrocytoma (SEGA), with incipient hydrocephalus in the left lateral ventricle.

From the medical history, the clinical examination and diagnostic procedures, the final diagnosis was established: TSC, bilateral obstructive kidney stones with left acute pyelonephritis and acute periurethritis. The diagnosis of tuberous sclerosis is supported by renal, cerebral and cutaneous modifications. The renal modifications consist in multiple kidney angiomyolipomas. The cerebral modifications consist of SEGA with incipient hydrocephalus on the left. The cutaneous modifications are represented by facial angiofibromas on the nasal wings, cheek and chin, hypopigmented plaques on the lower limbs, confetti-like lesions on the legs and small shagreen spots on the posterior hemithorax. The patient needed specific urological treatment for the bilateral obstructive kidney stones and was admitted in the urology clinical department, where the obstructive pyelonephritis and other associated diseases were treated. The patient received antibiotic treatment in concordance with the antibiogram (amikacin and cefoperazone/sulbactam), which led to a decrease in body temperature to normal values, and almost remitted the infectious process. After the complete remission of the infectious process, the patient underwent flexible ureteroscopy with laser lithotripsy, as this method was the most appropriate for the treatment of the patient. The genetic test revealed mutations in the TSC2 gene at chromosome 16p13.3. Her family did not want to perform a genetic test. So far, the patient has not had history of epileptic seizures, but over time has had intellectual disability (concentration and learning disorders), with mild psychomotor developmental deficits. Upon discharge, the patient began treatment with an m-TOR inhibitor (Everolimus 5 mg 1 tb/day).

## 3. Discussion

The major features for the clinical diagnostic criteria of TSC are: hypopigmented macules (>3, at least 5-mm in diameter), angiofibromas (>3) or fibrous cephalic plaque, ungual fibromas (>2), shagreen patch, multiple retinal hamartomas, cortical dysplasia, subependymal nodules, subependymal giant cell astrocytoma, cardiac rhabdomyoma, lymphangioleiomyomatosis (LAM) and angiomyolipomas (>2). Minor features are: “confetti” skin lesions, dental enamel pits (>3), intraoral fibromas (>2), retinal achromatic patches, multiple renal cysts, non-renal hamartomas. Two major features or one major feature with >2 minor features lead to the final diagnosis. Either one major feature or >2 minor features lead to a possible diagnosis [1].

SEGA can also be detected prenatally or at birth and has an incidence of 5–15% in TSC [4]. It is benign and typically slow-growing, but can cause obstructive hydrocephalus. An important source of mortality and morbidity in TSC are renal manifestations [5]. Angiomyolipomas are benign tumors composed of smooth muscle, vascular and fatty tissue [6]. They are observed most commonly in the kidneys of TSC patients, but can also occur in other organs. In 80% of TSC patients, fat-containing angiomyolipomas were observed [7]. Renal angiomyolipomas can cause serious issues with bleeding, because of their vascular nature, and can lead to the need for dialysis, and even renal transplantation [8]. In the general population, multiple renal cysts are not commonly observed [9], but can be seen in TSC patients who have a TSC1 or TSC2 mutation or as part of a contiguous gene deletion syndrome involving the TSC2 and PKD1 genes [6]. In 10–25% of TSC patients, liver angiomyolipomas are reported [10].

Differential diagnosis includes other causes of epilepsy, renal tumors, intellectual disability, rash and benign hamartoma syndromes. Vigabatrin is first-line treatment recommended for spasms in TSC. Steroids are a typical second line treatment, with sodium valproate as the third line option. Ketogenic diet is a treatment used in epilepsy related to TSC [11]. A symptomatic SEGA requires urgent surgery, which may include a ventriculoperitoneal shunt. Even if they cease spontaneously, bleeding renal angiomyolipomas have a high risk of re-bleed. Percutaneous embolization is the first choice of management. Smaller angiomyolipomas do not usually cause symptoms, but lesions larger than 4 cm in diameter are associated with an increased risk of serious hemorrhage [3,12].

Cysts greater than 4 cm in diameter are more likely to be symptomatic, and might present with flank pain or gross hematuria or as a tender mass [13]. Autosomal dominant polycystic kidney disease (ADPKD) will develop in patients with a contiguous deletion of the *PKD1* gene, which is associated with flank pain, hypertension, pyelonephritis and progressive renal failure [14].

## 4. Conclusions

Treatment should be organ specific, symptomatic and directed to improve the patient’s outcome and quality of life. Treatment and modern diagnostic imaging have improved both the life expectancy and the quality of life of patients with TSC. Quality of life and morbidity are largely determined by the neurologic manifestations, which include intellectual disability and seizures.

## Figures and Tables

**Figure 1 medicina-56-00369-f001:**
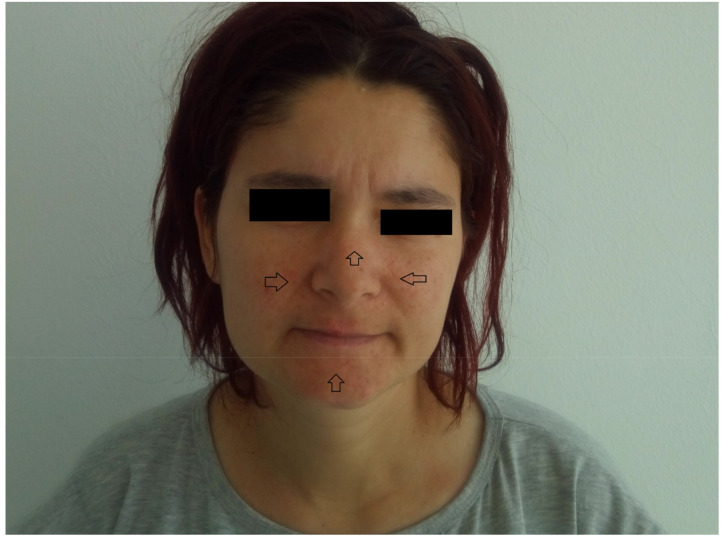
Facial angiofibromas on the nasal wings, cheek and chin.

**Figure 2 medicina-56-00369-f002:**
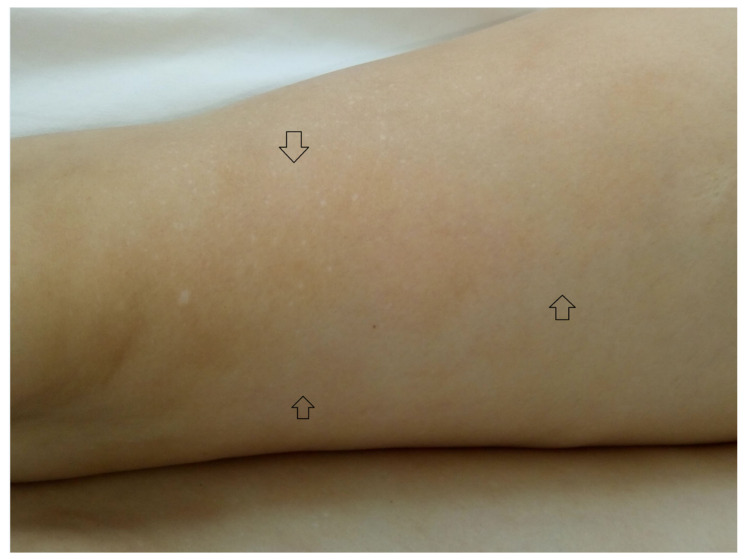
Hypopigmented plaques on the lower limbs.

**Figure 3 medicina-56-00369-f003:**
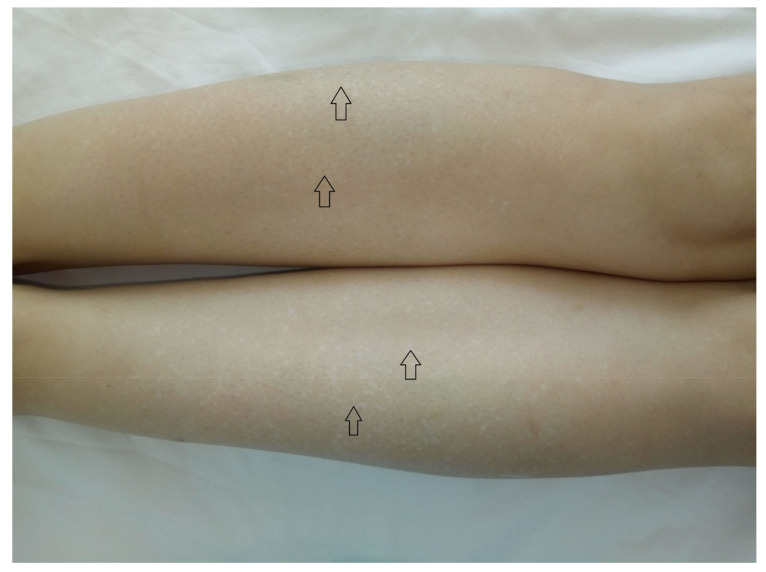
Confetti-like lesions on the legs.

**Figure 4 medicina-56-00369-f004:**
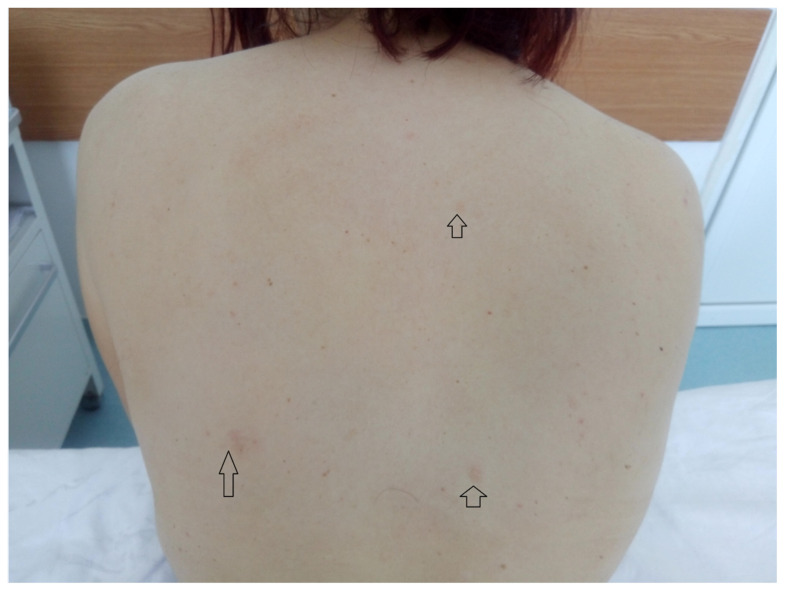
Small shagreen spots on the posterior hemithorax.

**Figure 5 medicina-56-00369-f005:**
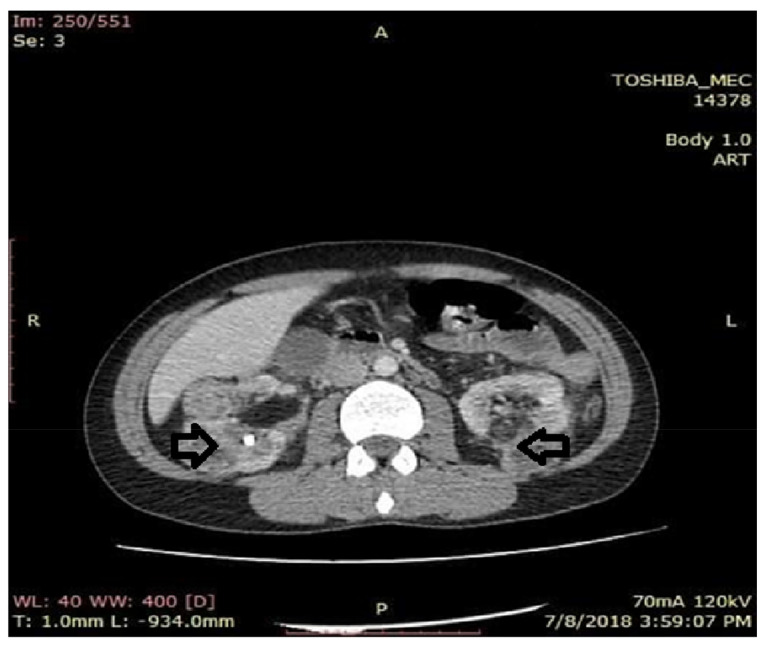
Axial computer tomography (CT) section with contrast substance. Kidneys with alteration of the physiological architecture with erasure of the cortico-medullary differentiation, by the presence on the entire bilateral renal surface of round oval formations, diffusely contoured with mixed component. The CT appearance pleads for angiomyolipomas.

**Figure 6 medicina-56-00369-f006:**
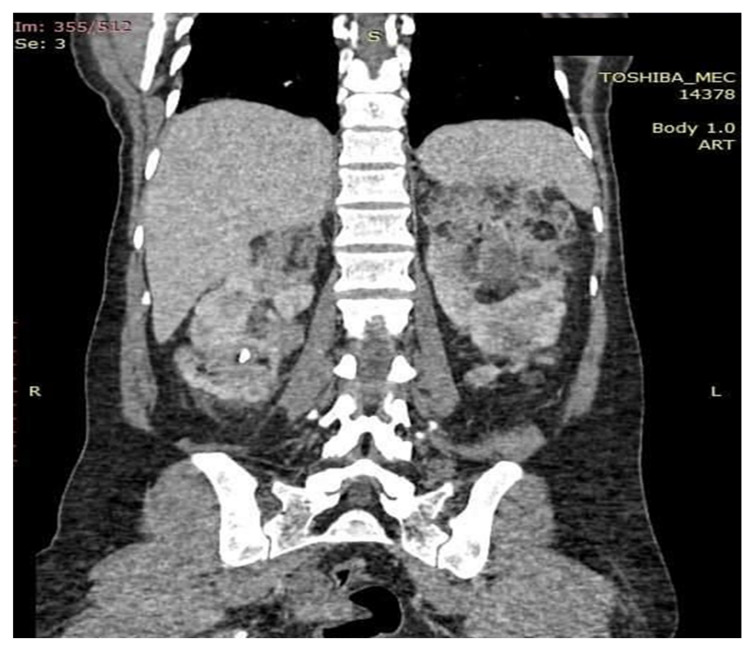
CT aspect of kidney modifications. Sagital CT section with contrast substance.

**Figure 7 medicina-56-00369-f007:**
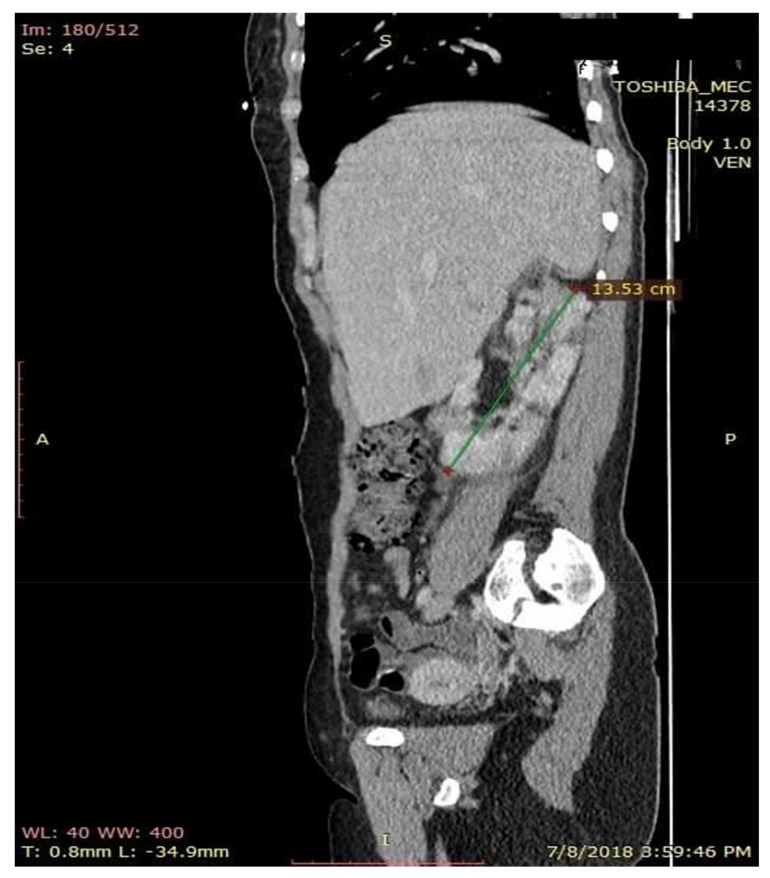
Coronal section examination CT. Kidneys enlarged in volume with the same changes in shape and structure by the presence of angiomyolipomatous formations that deform the renal contours without causing infiltration of proximity structures.

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
