# Peer review of "Management of a Patient with Tuberous Sclerosis with Urological Clinical Manifestations"

_medicina, 2020, doi:10.3390/medicina56080369_

Round 1
Reviewer 1 Report
The case report entitled “Management of a patient with Tuberous Sclerosis with urological clinical manifestations”, reported on an interesting topic regarding urological findings in Tuberous sclerosis, as first clinical signs.
The text and contents are understandable. However, there are some lack of informations that should be added.
There are several grammatical and typo errors. Therefore, an extensive English revision should be considered
Specific concerns:
Lines 27-28: The aim of this paper is to report an unusual such case in which urological clinical manifestations occurred and not neurological clinical manifestations: This sentence is not easy to read; please modify as follow “the aim of this paper is to report renal manifestations as first clinical signs suggestive of TSC diagnosis”
Line 47 page 2: Both genders and all ethnic groups are affected by TSC: please replase “are affected” with “could be affected”
Lines 49-50: Please put at line 47 before “both…” the sentence at lines 49-50:“TSC has an autosomal dominant mode of inheritance with almost complete penetrance, but variable expressivity
Line 54: please specify the significant pathological antecedents (including what?)
Line 58: Patient had a suggestive family history. For? Please add “for neurological diseases”
Line 60: Please add “At the admission” before “The patient presented
Line 67: To the physical examination, the skin has stigmas of TSC's disease. Please change the sentence: due to the presence of typical skin lesions TSC diagnosis was suspected.
Line 81: Neurological examinations showed reduced osteotendinous reflexes: which reflexes? Please add more information
Line 82-86: the sentence is too longer, please revise it and divide in more sentences
Line 89:please change suggesting for with “compatible with”
Line 90-91: “with kidney secretion and excretion conserved on the left side, but dilatory secretion of the right kidney…” please modify as follow: “ with dilatatory secretion in the right side and conserved secretion and excretion on the left side”. Then stop the sentence and start with a new sentence.
Line 91-92: please add this: There were also found images of a lithiasic component on the middle right caliceal branches (dimension of 7.1/6.5mm) and one of the left side with a smaller dimension, 2.5/2.5mm (figures 5, 6, 7).
Line 96: please correct “intraaortocav” with intraortocaval
Line 99: please add the localization of the subependymal giant cell astrocitoma
Line 100: left incipient hydrocephalus: what is the meaning? Maybe the author referred to the left lateral ventricle? Please add a figure of TC skull
In All the figures (1, 2, 3, 4, 5) please add some arrows to indicate the specific target described
Figure 5 to 7: please add some informations in the legend with arrows
Has been performed the genetic test to investigate mutations in either the TSC2 gene at chromosome 16p13.3, or the TSC1 gene at chromosome 9q34. It is important to note as she started treatment with an m-TOR inhibitor.
Considering the family history of tumors, has been a genetic test performed in her family?
Was SEGA treated with surgery as an incipient hydrocephalus was reported. Did a neurologist perform the neurological examination? Did the woman have intellectual disability? Was normal her psychomotor development? Did she have any history of epileptic seizures? Please add these informations. In addition please report some information regarding cardiologic findings using echocardiogram that cuold be present in TSC.
Line 134: Please delete “Subependymal giant cell astrocytoma” and report only “SEGA”
Line 140: In the kidney angiomyolipomas; please change “in the kidney” with “renal”
Line 144: plase put “liver angiomyolipomas” before “are reported”
Line 147: please add also ketogenic diet between treatment used in epilepsy related to TSC. Please add this reference
Youn SE, Park S, Kim SH, Lee JS, Kim HD, Kang HC Long-term outcomes of ketogenic diet in patients with tuberous sclerosis complex-derived epilepsy. . Epilepsy Res. 2020 Aug;164:106348. doi: 10.1016/j.eplepsyres.2020.106348.
Author Response
Dear reviewer,
Thank you very much for your recommendations and suggestions in your review report.
We make an extensive English revision and we hope that we correct all the errors of language and style. We highlighted the modifications of spelling and grammar.
We try to resolve all your specific recommendations and suggestions.
Best regards,
Vlad Padureanu
Reviewer 2 Report
Review of MS#845634 [Management of a patient with Tuberous Sclerosis with urological clinical manifestations] by Vlad Padureanu et. al.
This manuscript presents very interesting data on a TSC patent case with renal calculus but without renal angiomyolipomas. The paper is well-writtened, however, the figures should be improved to high quality (e.g. Figure 2 Hypomelanotic plaques; Figure 3 Confetti-like lesions; Figure 4 Shagreen spots).
This Ms is acceptable for publication after minor revision.
Author Response
Dear reviewer,
Thank you very much for your review report. We improved to high quality the figures in the manuscript.
Best regards,
Vlad Padureanu
Round 2
Reviewer 1 Report
Only few specific concerns should be considered:
Lines 28: please add a poin at the end of the sentence
Line 66: please delete a poin at the end of the sentence
In the figures 1, 2, 3, 4, 5 please add some arrows or some points to indicate the target of interest
Author Response
Dear reviewer,
Thank you very much for your review report.
Line 28: We add a point at the end of the sentence
Line 66: We delete a point at the end of the sentence
In the figures 1, 2, 3, 4, 5 we add some arrows to indicate the target of interest
Best regards,
Vlad Padureanu